# Defining the Most Effective Patient Blood Management Combined with Tranexamic Acid Regime in Primary Uncemented Total Hip Replacement Surgery [note 1]

**DOI:** 10.3390/jcm9061952

**Published:** 2020-06-22

**Authors:** Hanna Pérez-Chrzanowska, Norma G. Padilla-Eguiluz, Enrique Gómez-Barrena

**Affiliations:** 1Department of Anaesthesiology, Hospital Universitario “La Paz”, 28046 Madrid, Spain; hannaperez@hotmail.com; 2Department of Orthopaedics, Hospital Universitario “La Paz”, IdiPAZ, and Autónoma University, 28046 Madrid, Spain; normapadilla@gmail.com

**Keywords:** uncemented total hip replacement, patient blood management, tranexamic acid, optimal protocol, bloodless medicine

## Abstract

The application of patient blood management (PBM) combined with tranexamic acid administration (TXA) results in decreased total blood loss volume (TVB) and transfusions in total hip replacements (THRs). Dosages, timing, and routes of administration of TXA are still under debate as all these aspects, as well as interpatient variations, may affect the efficacy of the protocol. This study aims to examine the effectiveness of timing and route of administration of TXA in combination with PBM by reducing the TBV following THR surgery. Consecutive primary uncemented THRs operated by a single surgical and anaesthetic team had the data prospectively collected and then retrospectively studied. Five treatment groups were formed, reflecting the progressive evolution of our protocol. Group 1 included patients managed with PBM alone (preoperative erythrocyte mass optimisation to at least 14 g/dL haemoglobin (Hb), hypotensive spinal anaesthesia and restrictive red blood cell transfusion criteria). Group 2 included patients with PBM and topical 3 g TXA diluted in normal saline to a total volume of 50 mL. Group 3 were patients with PBM and an IV dose of 20 mg/kg TXA at induction, followed by 20 mg/kg TXA as a continuous infusion for the duration of the operation. Group 4 consisted of patients managed as per Group 3 plus another 20 mg/kg TXA at three-hour post-procedure. Group 5 (combined): PBM and IV TXA as per Group 4 and topical TXA as per Group 2. A generalised linear model with the treatment group as an independent variable was modelled, using TBV as the dependent variable. The transfusion rate for all groups was 0%. TBV at 24 h, oscillated from 613.5 ± 337.63 mL in Group 1 to 376.29 ± 135.0 mL in Group 5. TBV at 48 h oscillated from 738.3 ± 367.3 mL (PBM group) to 434 ± 155.2 mL (PBM + combined group). The multivariate regression model confirmed a significant decrease of TBV in all groups with TXA compared with the PBM-only group. Overweight and preoperative Hb were confirmed to significantly influence TBV. The optimal regime to achieve the least TBV and a transfusion rate of 0% requires PBM and one loading 20 mg/kg dose of TXA, followed by continuous infusion of 20 mg/kg for the duration of the operation in uncemented THRs. Additional doses of TXA did not add a clear benefit.

## 1. Introduction

In 2006, the Anaesthesiology and Orthopedics services at the Hospital Universitario La Paz-Hospital Cantoblanco conducted an internal audit of blood transfusions in uncemented primary total hip replacements (THRs), observing a 51.2% rate of transfusions. In 2007, we introduced an evolving protocol of a bloodless medicine program, applying the three pillars of patient blood management (PBM) to THR patients. When the PBM was consolidated, we administrated TXA in various regimes, following the magnificent clinical results that achieved a 0% rate of blood transfusion with the application of similar protocols in total knee replacements (TKRs) [1].

According to blood bank data, the transfusion rate in Cantoblanco Hospital for THRs changed from 51.2%, before the introduction of the PBM protocol, to a rate of 0% registered in 2007 through to 2015, when, due to structural changes to operating room programming, the Cantoblanco Hospital stopped receiving THR patients. Although the decreased transfusion rate was clearly identified as an important achievement of the PBM program, the information related to the component of the reduction of blood loss with each step of the evolution of TXA regimes, is unknown. Multiple literature references and meta-analysis have been published [2,3,4,5] about the use of TXA in this field, but the ideal regime eludes us.

Alshryda et al. [2] performed a systematic review and meta-analysis of the topical administration of TXA in THRs and TKRs. They found a significant reduction of blood transfusions in both procedures. The rate of thromboembolic events was similar to that found with placebo. They ventured that topical administration is superior to the IV route after an indirect comparison and concluded that further research is required to find the optimal dose for the topical use of TXA.

Sukeik et al. [3] performed a systematic review and meta-analysis of published randomised controlled trials evaluating the efficacy of IV TXA in reducing blood loss and transfusion in THRs. They found that its use reduced the intraoperative and postoperative bleeding, as well as total blood volume loss (TBV), leading to a significant reduction of transfusions. They also found no significant differences in various complications (deep venous thrombosis (DVT), pulmonary embolism (PE), infection, and others) between the control and study groups.

Wei and Biaofang [4] described a study of topical versus IV TXA administration in THRs, comparing the results. They concluded that both methods are equally effective in reducing blood loss and transfusion rates without increased complications.

Wei and Liu [5] reported a meta-analysis assessing total blood loss, the incidences of DVT and PE, and the number of patients requiring at least one unit of red blood cells following THR and TKR, with TXA irrespective of the route of administration. Their results suggested that TXA significantly reduces blood loss and the need for allogenic blood transfusion without apparent increased risk of DVT or PE.

We hypothesize that the application of TXA combined with PBM for uncemented THR would reduce the TBV compared with PBM alone. The primary objective of this study is to compare the uncemented THR postoperative blood loss at 24 and 48 h in the different PBM and TXA administration groups. Secondary objectives investigate the influence of dependent variables in TBV.

## 2. Patients AND Methods

A retrospectively analysed cohort study of prospectively collected data from uncemented THR patients operated at the Hospital Universitario La Paz-Cantoblanco from January of 2007 through to December of 2015 was performed. The patients were operated on in the lateral position, and the implants were not cemented. Two drains at atmospheric pressure were left in situ for 48 h. Patients fell chronologically into one of five groups. Group 1 (PBM) reflected Stage 1 of our program and included patients with red blood cell mass optimization up to at least 14 gr/dLHb, controlled hypotension spinal anaesthesia coadjuvated by IV midazolam and a continuous infusion of propofol sedation intraoperatively, a high volume/low concentration infiltration of 200 mg ropivacaine, 0.5 mg adrenaline, 100 mg tobramycin, 10 mg morphine, 4 + 4 mg betametasone (Celestone^®^) was infiltrated into the tissues surrounding the operated area before wound closure for the coadjuvation of postoperative analgesia and to decrease local blood loss, together with restrictive transfusion criteria (transfusion if Hb under 7 gr/dL, elective transfusion if symptomatic and between 7 and 9 gr/dL, no transfusion if over 9 gr/dL and asymptomatic) throughout the hospital stay. Group 2 (PBM + Topical) defined as per Group 1 plus a topical intra-articular administration of 3 gr of TXA in 20 mL of normal saline (total volume of 50 mL). This group was created for patients with IV TXA contraindication (recent urogenital bleeding, pulmonary embolism, myocardial infarction, recent coronary artery stenting or dilatation, cerebrovascular thrombosis, epilepsy). Group 3 (PBM + IV) was followed in time and was managed as per Group 1 plus a loading dose of IV TXA at 20 mg/kg, followed by 20 mg/kg IV TXA in continuous infusion for the duration of the operation. Next, we implemented Group 4 (PBM + 2IV) as per Group 3 plus another dose of 20 mg/kg of IV TXA 3 h post-procedure. Group 5 (PBM + Combined) was as per Group 4 plus 3 gr of TXA intraarticularly diluted to 50 mL total volume with saline. The study was conducted in accordance with the Declaration of Helsinki, and the protocol was approved by the Ethics Committee of Hospital Universitario La Paz under the registration number PI-3243.

All consecutive patients surgically treated for uncemented THR by the same surgical team from 2007 to 2015 were included in the study, except for those that met at least one of the following exclusion criteria: revision hip arthroplasty, discharge before 48 h postoperative, incomplete anaesthetic graphics, incomplete laboratory data pre- and postoperative (at 48 h). Exposure variables correspond with the treatment assignation, and total follow up was of 48 hr postoperative.

The following data were collected: group of treatment, age (years); gender (male/female”); weight (kg); date of the operation, length of surgery (min); side of the prosthesis (left, right); anaesthetic risk status according to the American Society of Anesthesiologists (ASA) stratification (I, II, III); pretreatment with IV iron and erythropoietin [Fe: Iron; EPO: erythropoietin; Fe + EPO: both; N: none]; anaesthetic technique [spinal, controlled hypotension]; intraoperative blood loss estimation (mL); blood loss into drains at 3 h postop and 24 h (mL); Hb preoperative and postoperative at 3, 24, and 48 h (g/dL); Hct preoperative and postoperative at 3, 24, and 48 h (%); length of hospital stay [days]. To estimate the body mass index (BMI), missing height was obtained from the demographics database of the Spanish National Statistics Institute (2015) by sex and age categories: height = 1.75 m if males under 55 years old (yo); height = 1.67 m if males over 54 yo; height = 1.62 m if females under 55 yo; height = 1.54 m if females over 54 yo.

A total of 159 hips were identified, but finally, 153 hips (133 patients) were included in the study (4 revisions and 2 cemented were excluded, no exclusion was related to discharge before 48 h). Of these, 29 hips (19%) were allocated to the PBM group, 18 (12%) to the PBM + Topical group, 42 (27%) to the PBM + IV group, 41 (27%) to the PBM + 2IV group, and 23 (15%) to the PBM + Combined group.

The mean age of the treated patients was 67 ± 12 years, 47% were females (*n* = 72), 44% were overweight (BMI 25 to 29), and 67% were classified as ASA II. According to the baseline characteristics (Table 1), no differences among groups were found except for the age, suggesting that patients of the PBM + Combined group were older (95% CI of age: 68.1–76.1 years) than participants of PBM + IV group (95% CI of age: 62.3–68.1 years). A difference among groups is observed in the obesity category (Fisher’s exact, *p* = 0.022) when reshaping it in two groups (obesity vs. no-obesity), with a higher proportion of obese patients in the PBM + 2IV group.

No differences were observed on preoperative laboratory parameters; the mean haemoglobin concentration was 14.65 ± 1.57 g/dL, and the mean haematocrit was 44.38 ± 3.69%.

The principal outcome was to compare the TBV at 48 h postoperative among the treatment groups. TBV was calculated multiplying the total circulating blood volume (CBV) by the difference between postoperative Hct at 48 h with basal Hct [6,7]:

TBV = CBV * (Hct _48 h post-operative_ − Hct _pre-operative_), where CBV = (70/100 * weight in kg).

For secondary outcomes, the intraoperative blood loss and the postoperative drop of Hb and Hct (at 24 and 48 h) from basal levels were compared among the treated groups.

The general characteristics and primary and secondary outcome variables were compared with a confidence of 95% among the groups of treatment using Wilks’ lambda if homoscedasticity, Wald’s test (James’ approximation) if heteroscedasticity, or Fisher’s exact test for categorical variables [8,9]. To determine which group was different, the modification of the Kruskal-Wallis test was used or the Wald test after multivariate regression [10]. A generalised linear model was conducted using Gaussian variance and identity link functions due to the assumptions for a multivariate ordinary least squares regression not being met. The dependent variable was the TBV loss at 48 h after surgery. The independent variables were the assigned group of treatment adjusted for ASA, age, sex, BMI, and preoperative Hb concentration (1 > 14, 2 = 12 to 14, 3 < 12). Power estimations (β) of the comparisons were conducted. Data were analysed using STATA software version 11 (StataCorp. 2009; Stata Statistical Software: Release 11; College Station, TX: StataCorp LP).

## 3. Results

Primary and secondary outcome data are shown in Table 2. The drop of Hb (Figure 1a) and Hct (Figure 1b) did not show statistically significant differences among the groups with TXA, at 24 and 48 h postoperative, but the drop was significantly higher in the PBM group (*p* < 0.01 for both). Differences were observed in the mean length of hospital stay (LOS) among groups, from 2 to 10 days. When categorised, the shorter LOS of 2 days was found in the PBM + 2IVgroup and the PBM + Combined group in more than 70% of patients. The second more common LOS was from 6 to 10 days, in which more than 60% of patients of the PBM group, the PBM + Topical group, and the PBM + IV group were found.

A similar volume of blood loss was found among the groups that received the PBM program and at least one dose of TXA (Figure 2), at 24 and 48 h postop. These volumes were significantly lower than the one found in the PBM group (*p* < 0.01) except for the PBM + Topical group, where its mean was lower but failed to reach significance.

After controlling the TBV loss at 48 h postop with the associated covariates in the GLM model (Table 3), no differences were found among all the groups that received the PBM program and at least one dose of TXA against the PBM group (Figure 3). Compared to PBM, the difference of predicted TBV varied from −238 mL, if treated in the PBM + Topical group, to −295 mL, if treated in the PMB + Combined group.

## 4. Discussion 

The Royal College of Physicians of Great Britain audit in 2007 showed a wide variation in transfusions of THRs, with an average of 25% (range 22–97%) [11]. Many strategies are being implemented to decrease it, as blood transfusion is beneficial and essential in some patients, but it is not without high inherent risks. The various strategies implemented can be resumed as pre-, intra-, and postoperative. Among the preoperative strategies, optimisation of patients’ Hb/Hct levels is of primary importance; however, different groups report different levels as their objective [12,13,14]. Our team performed a preliminary audit of THR patients operated on at the Hospital La Paz-Cantoblanco in 2006 and found that without taking any steps to reduce blood loss, our patients lost a median of 150 mL (range 150–600 mL) intraoperatively and a median 1000 mL (range 100–2000 mL) in the drains at 3 h. The mean Hb drop at 48 h was 4 g/dL (51.2% of patients were transfused). From this data, we extrapolated that to avoid the risk of transfusion (Hb transfusion threshold of 10 g/dL before the implementation of restrictive transfusion criteria), the patients had to enter the operating room with a Hb of a least 14 g/dL. This became our aim as we introduced PBM to our practice. At the time of our internal audit in 2006, intraoperative cell salvage was deemed not to be cost-effective, and we did not have access to reinfusion drains. As soon as the logistics permitted, we opened up a facility to administer IV iron (Fe) and subcutaneous (sc) EPO, with the appropriate patient monitoring in our postoperative recovery unit first thing in the morning before the arrival of the patients from the operating theatre. At the same time, we introduced the principles of PBM, adhering to restrictive blood transfusion thresholds (RBC transfusion for all patients with a Hb of 7 or less g/dL, no transfusion above 10 g/dL, and transfusion at the discretion of the anaesthesiologist and surgeon in patients who were symptomatic or had comorbidities with a Hb of 7–9 g/dL). It is important to consider the degree of postoperative anaemia in patients who do not qualify for RBC transfusion, as it may hinder the speed of recovery. We included postoperative IV Fe supplementation for anaemic patients who did not receive a transfusion, but its discussion is beyond the scope of the present study.

We implemented controlled hypotension with an acute hypervolemic haemodilution to compensate for the vasodilatory effects of the anaesthetic rather than using vasoconstrictors such as efedrin to maintain the desired blood pressure (BP) at 20% less than the patient´s resting mean BP (at the discretion of the anaesthesiologist) during the operation as a strategy to minimise intraoperative blood loss. As all the patients´ comorbidities were optimised in the preoperative period in the preanaesthetic clinic, we had no contraindications to implementing this technique. Later, we also started applying TXA.

Thus, we formed the groups of patients. Group 1 were patients who received IV Fe to reach the prerequisite of Hb 14 g/dL. All patients were operated on with a spinal hypotensive anaesthetic, with 15 mg of hypobaric bupivacaine and 10 mcg of fentanyl after premedication with 2–3 mg of IV midazolam (dose adjusted at the extremes of height, weight, and age), followed by continuous infusion of propofol to maintain adequate spontaneous ventilation, fluid administration, and haemodilution, at the discretion of the anaesthesiologist, to maintain a mean BP of 20% below resting BP. This level of hypotension can be considered as moderate and was well tolerated by all our patients. Apart from meticulous haemostasis and the administration of antifibrinolytics, there were no other methods to decrease the intraoperative blood loss. It can be argued that depending on the cost of intraoperative cell salvage, it may become cost-effective if it prevents the transfusion of a unit of RBC per patient. This was not the case for our cohorts.

In our experience with TKR, IV TXA was instrumental in achieving a 0% transfusion rate [1]. Although the pathophysiology of hyperfibrinolysis is different in TKR with ischemia (where the exsanguination tourniquet leads to intense liberation of local anaerobic metabolism byproducts and hyperfibrinolisis), we extrapolated from the findings of the CRASH-2 trial [15,16] and presumed some degree of hyperfibinolysis to also be present in THR due to the large muscles and long bone trauma. Different dosage schemes were appearing in the literature, and we started implementing them progressively, beginning with a 20 mg/kg loading dose on the induction of anaesthesia, followed by a continuous infusion of the same dose for the duration of the operation (Group 3). It was immediately clinically obvious that the addition of IV TXA worked very well and we passed from a mean 950 +/− 589.4 mL of intraoperative blood loss to a mean of 253.1 +/− 137.7 mL. A similar drastic drop in drain yield was found from a mean of 172.2 +/− 168.5 mL at 3 h to a mean of 71.3 +/− 90.3 mL, and a mean of 365.5 +/− 262.3 mL at 24 h to a mean of 296.9 +/− 189 mL. With these results, we discontinued the use of wound drains and were able to introduce an enhanced recovery program and shorten the LOS from 9.4 ± 3.1 days to a mean of 2.3 ± 0.8 days. This is the reason why we have no drain yield results for Groups 3, 4 and 5 (we did not report it in the results), and why the Hb/Hct of these patients in these groups were not followed up for 5 days. As we discontinued the use of drains and thought that clinically significant hyperfibrinolysis continued for at least 8 hrs postop, we added an additional dose of 20 mg/kg of TXA in the recovery unit 3 hrs after the end of the operation (Groups 4 and 5). The pathophysiology of hyperfibrinolysis in THR is different from that of TKR, and, as seen in the results, there was no significant difference (statistically or clinically) with this added dose. In addition, as we had augmented benefits to postop blood loss with a topical application of TXA added on to the IV regime in TKRs, we created Group 5 to see if we could obtain a similar improvement in THRs. The results of this study show that this is not so, and there were no clinically or statistically relevant benefits of the addition of topical TXA to the IV TXA regime, although we do not have the power to detect equivalences with Group 3 results. With the intention of benefiting patients with a contraindication for IV TXA, we created Group 2 of the patients, where the whole PBM protocol was applied and TXA was administered topically in quantities that do not reach significant systemic plasma levels.

We detected no complications in any of the groups. However, this study was not designed to describe the possible adverse outcomes. The safety of IV and topical TXA administration has been sufficiently addressed in the literature [17,18].

It is beyond the scope of this study to evaluate the economic benefits of the incorporation of our protocol compared to the 2006 pre-PBM and TXA administration scenario. However, the savings in direct costs of the administration of RBC and avoiding their complications, the shortening of hospital stays, and enhanced patient recovery perfectly cover the relatively small costs of IV Fe +/− sc EPO administration to a very minor population [19].

Literature searches on blood-saving in THRs offer different degrees of total blood loss and transfusion rate decrease, but we have not found any published results achieving a 0% transfusion rate [17,20]. This is an extraordinary result, which has remained unaltered during the eight years of the duration of this study. Personalised PBM and carefully individualised anaesthetic technique may be relevant and difficult to standardise. Although there are many TXA dosing regimes described in the literature, there is none which could be considered as optimal and definitive, due to the lack of a large cohort study.

We have found that our protocols were easy to implement and highly reproducible by other teams, always with a clear emphasis on PBM. These rely on close multidisciplinary cooperation of surgeons and anaesthesiologists during the whole duration of the patients´ hospital stay. Implementing a PBM strategy is vital for the success of any blood-saving program. Each institution or team should perform an audit of the Hb/Hct drop after their THRs and extrapolate the optimal Hb level the proposed patients should reach before the operation to safely achieve a margin for perioperative blood loss and complications. The results of this study suggest that the PBM protocol itself can save more than 300 mL of blood, as can be seen in the comparison of total blood loss in Table 4, although a more structured study is needed to confirm it. Interestingly, this is observed by comparing our PBM group to the placebo arms used in published studies [4,21,22,23,24,25,26,27,28].

Data from GLM regression suggest that obesity contributes to blood loss, with higher volumes of TBV between the PBM + IV, PBM + 2IV, and PBM + Combined groups at 48 h postop that could make us consider the use of two doses of TXA or an alternative dose of TXA, which could be utilized in addition to more frequent dosing in this specific group of patients. Patients with contraindication to the use of TXA should follow PBM to reduce the risk of blood loss and transfusion.

Among the observed limitations, we must recognise that the study could have a selection bias because our health care system allocates the population to a hospital by influence area; although no demographic or major variability is expected in comparison with the rest of the population of Madrid, the results of this study may not be fully generalized. Similarly, the lack of randomisation could lead to potential confounders. The number of included cases is not high, and this could lead to potential underpowering of the study, specifically to identify equivalence among TXA groups; however, it may serve as a pilot study. Patients of PBM and PBM + Topical TXA groups are probably different from the rest due to comorbidities (prompting avoidance of IV TXA), although this is not observed by the severity variable most frequently used (ASA classification). However, the fact that all cases were performed by the same anaesthetic and surgical team decreases the potential anaesthetic and surgical variability, which might also impact the perioperative blood loss.

## 5. Conclusions

According to our results, PBM plus IV TXA is the most effective regime to minimise intra- and postoperative-bleeding and should be used in all patients without contraindications. Doses of 20 mg/kg TXA at induction, followed by 20 mg/kg infusion for the duration of the procedure, are sufficient to consolidate a 0% transfusion rate. In patients with IV TXA contraindications, the preoperative Hb optimisation and PBM becomes vital. Although in our study, the application of topical TXA did not confer statistically significant difference to the results, the clinical importance of the diminished TBV loss (200.8 mL difference between the PBM and PBM + Topical groups) leads us to advocate its use as safe and effective. A large multicentric study of PBM and TXA in primary uncemented THRs is needed to fully confirm the reproducibility of our results.

## Figures and Tables

**Figure 1 jcm-09-01952-f001:**
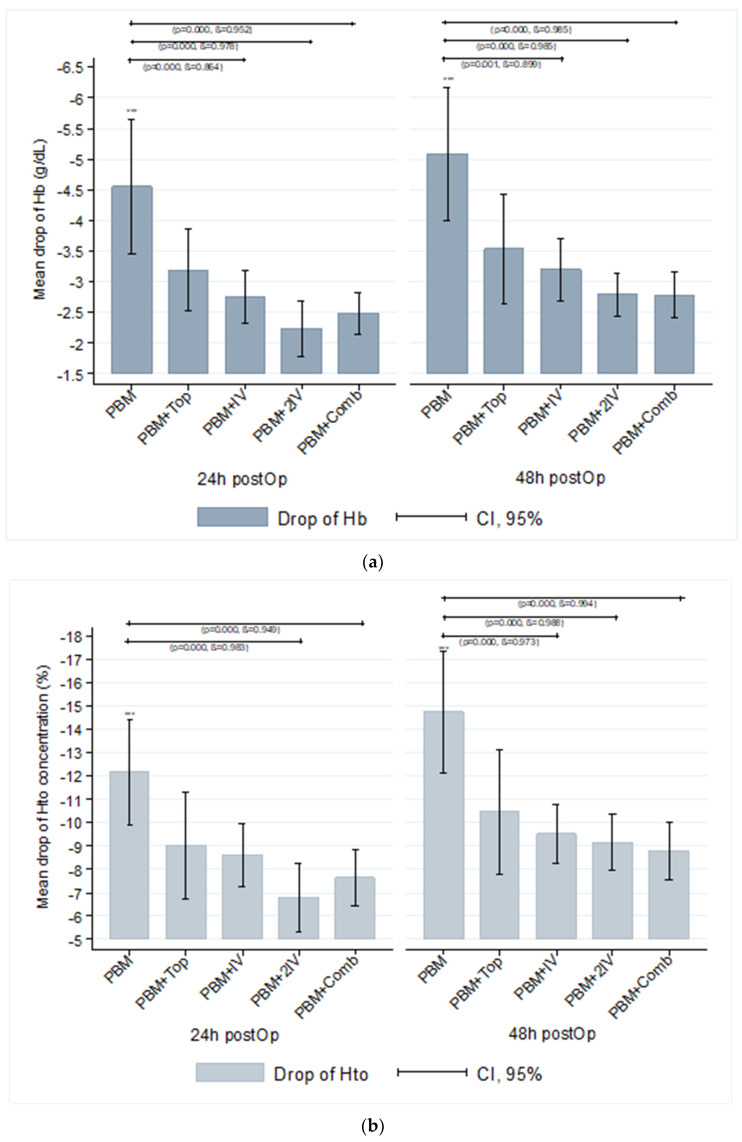
Laboratory differences at 24 and 48 h postoperative by group of treatment. (**a**) Drop of Haemoglobin concentration (g/dL); (**b**) Drop of Haematocrit concentration (%). *** The mean drop of haemoglobin concentration and haematocrit concentration shows differences (Kruskal–Wallis test) between the PBM group and other TXA groups. No differences were found in comparisons between each TXA group with other TXA groups (*p* > 0.05, ß < 50%).

**Figure 2 jcm-09-01952-f002:**
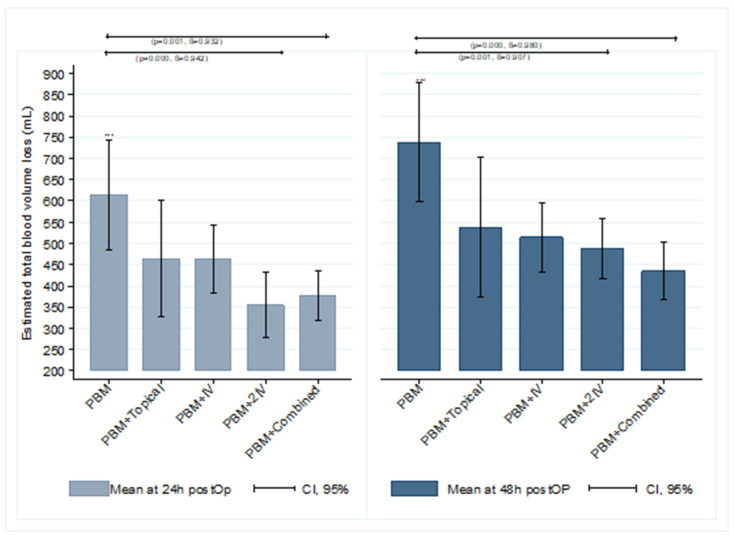
Total blood volume loss at 24 and 48 h postoperative by group of treatment. *** The mean estimated blood loss volume shows differences (Kruskal–Wallis test) between the PBM group and PBM + 2IV and PBM + Combined groups at 24 and 48 h. No differences were found in comparisons between each TXA group with other TXA groups (*p* > 0.05, ß < 50%).

**Figure 3 jcm-09-01952-f003:**
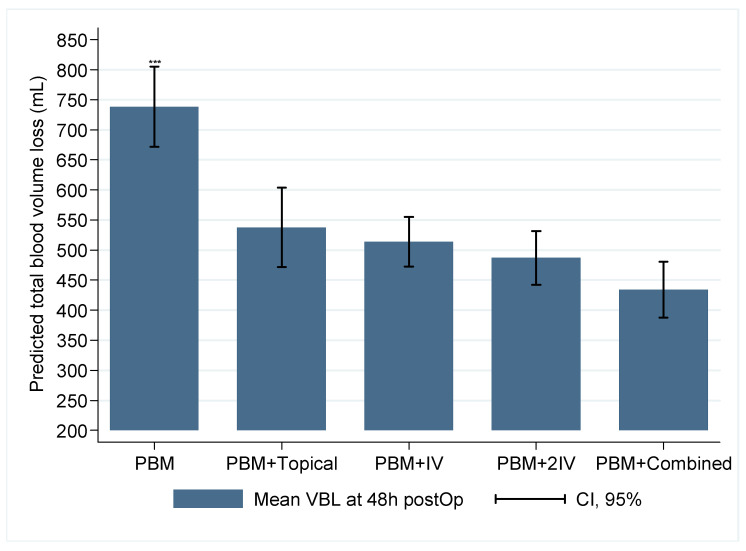
Comparison of predicted total blood volume loss at 48 h postop among groups of treatment, adjusted by the rest of the variables in the model. *** Difference (Wald’s test) between the PBM group with each TXA group (*p* < 0.001, ß > 95%). No differences were found in comparisons between PBM + Topical/PBM + IV (*p* = 0.971, ß = 99%), PBM + Topical/PBM + 2IV (*p* = 0.260, ß = 26%), PBM + Topical/PBM + Combined (*p* = 0.766, ß = 78%), PBM + IV/PBM + 2IV (*p* = 0.268, ß = 14%), PBM + IV/PBM + Combined (*p* = 0.244, ß = 74%), and PBM + 2IV/PBM + Combined (*p* = 0.975, ß = 31%).

**Table 1 jcm-09-01952-t001:** Demographics and general characteristics of patients by group of treatment.

Variables	Group 1 PBM (*n* = 29)		Group 2 PBM + Topical (*n* = 18)		Group 3 PBM + IV (*n* = 42)		Group 4 PBM + 2IV (*n* = 41)		Group 5 PBM + Combined (*n* = 23)		*p*-Value *
	Mean ± sd	(min–max)	Mean ± sd	(min–max)	Mean ± sd	(min–max)	Mean ± sd	(min–max)	Mean ± sd	(min–max)	
	*n*	(%)	*n*	(%)	*n*	(%)	*n*	(%)	*n*	(%)	
Age (yo)	64 ± 17	(18–83)	68 ± 13	(33–88)	65 ± 9	(49–82)	70 ± 11	(43–91)	72 ± 9	(48–87)	**0.028**
Female sex	15	(52%)	8	(44%)	16	(38%)	22	(53%)	11	(48%)	0.668
Weight (kg)	71 ± 20	(40–156)	71 ± 14	(50–101)	75 ± 15	(42–110)	76 ± 12	(50–108)	70 ± 9	(50–84)	0.247
Body Mass Index (BMI)	27 ± 7	(15–56)	27 ± 4	(21–36)	28 ± 5	(18–39)	30 ± 5	(20–44)	27 ± 3	(21–35)	0.171
Obesity category:											
Healthy weight	11	(38%)	6	(33%)	11	(26%)	7	(17%)	5	(22%)	
Overweight	10	(34%)	10	(56%)	18	(43%)	15	(37%)	15	(65%)	0.055
Obese	8	(28%)	2	(11%)	13	(31%)	19	(46%)	3	(13%)	
ASA class	2.2 ± 0.6	(1–3)	2.0 ± 0.7	(1–3)	2.2 ± 0.5	(1–3)	2.2 ± 0.4	(1–3)	2.2 ± 0.3	(2–3)	0.965
I	4	(14%)	4	(22%)	3	(7%)	1	(2%)	0	(%)	0.123
II	16	(55%)	9	(50%)	28	(67%)	31	(76%)	19	(83%)	
III	9	(31%)	5	(28%)	11	(26%)	9	(22%)	4	(17%)	
Left side	10	(35%)	9	(50%)	19	(45%)	14	(34%)	12	(52%)	0.309
Preoperative intake of Iron/EPO											
None	27	(93%)	15	(83%)	34	(81%)	28	(67%)	18	(78%)	0.087
Iron	2	(7%)	2	(11%)	5	(12%)	12	(29%)	2	(9%)	
Iron-EPO	0	(0%)	1	(6%)	3	(7%)	1	(2%)	3	(13%)	

* For continuous variables: Wald ji2 test (James’ approximation), allowing heteroscedasticity. For categorical variables: Fisher’s exact test. Bold if statistically significant at 95% of confidence.

**Table 2 jcm-09-01952-t002:** Drop of haemoglobin and haematocrit and blood loss variables by group of treatment.

Variables	Group 1 PBM (*n* = 29)		Group 2 PBM + Topical (*n* = 18)		Group 3 PBM + IV (*n* = 42)		Group 4 PBM + 2IV (*n* = 41)		Group 5 PBM + Combined (*n* = 23)		*p*-Value
	Mean	(CI, 95%)	Mean	(CI, 95%)	Mean	(CI, 95%)	Mean	(CI, 95%)	Mean	(CI, 95%)	
**Haemoglobin concentration (g/dL)**											
PreOp	15.0	(14.1–15.9)	14.9	(14.4–15.5)	14.6	(14.1–15.0)	14.4	(13.9–14.7)	14.5	(14.0–14.9)	0.371 *
24 h postOp	10.5	(9.8–11.1)	11.7	(11.1–12.4)	11.9	(11.4–12.3)	12.0	(11.5–12.6)	12.0	(11.4–12.6)	**0.000** **
48 h postOp	9.9	(9.3–10.6)	11.4	(10.5–12.5)	11.4	(10.9–11.9)	11.5	(11.1–12.0)	11.7	(11.2–12.2)	**0.000** **
**Haematocrit concentration (%)**											
PreOp	44.4	(12.8–45.9)	45.3	(43.8–46.8)	44.3	(43.1–45.4)	44.2	(42.9–45.5)	44.0	(42.6–45.3)	0.840 *
24 h postOp	32.2	(30.2–34.5)	36.3	(34.0–38.5)	35.7	(34.2–37.1)	37.4	(35.9–38.9)	36.4	(34.6–38.1)	**0.000** **
48 h postOp	29.6	(27.1–32.1)	34.8	(32.1–37.5)	34.8	(33.3–36.2)	35.0	(33.7–36.3)	35.2	(33.6–36.8)	**0.004** *
**Blood loss**											
Intraoperative blood loss	950.0	(711–1188)	286.1	(200–371)	253.1	(209–296)	252.4	(207–297)	252.2	(186–318)	**0.000** *
Total blood loss at 24 h postOp	613.5	(485– 741)	464.2	(328–599)	462.6	(381–543)	354.1	(277–431)	376.3	(317–434)	**0.005** *
Total blood loss at 48 h postOp	738.3	(598–878)	537.5	(373–701)	513.7	(431–598)	487.0	(417–556)	434.2	(367–501)	**0.005** *
**Length of hospital Stay**	9.5	(8.3–10.6)	6.5	(5.6–7.4)	8.0	(7.2–8.9)	2.7	(2.3–3.0)	2.4	(2.0–2.7)	**0.000** *
**LOS category *n* (%):**											**0.000 ^¥^**
2 days	0	-	0	-	0	-	28	(68.3%)	18	(78.3%)	
3–5 days	1	(3.4%)	6	(33.3%)	2	(4.7%)	11	(26.8)	5	(21.7%)	
6–10 days	21	(72.4%)	11	(61.1%)	33	(78.6%)	2	(4.9%)	0	-	
>10 days	7	(24.1%)	1	(5.6%)	7	(16.7%)	0	-	0	-	

PreOp: preooperative; postOp: postoperative. * Wald ji2 test (James’ approximation), allowing heteroscedasticity; ** Wilks’ lambda test. **^¥^** Fisher Exact test. Bold if statistically significant at 95% of confidence.

**Table 3 jcm-09-01952-t003:** Generalised linear model analysis of the association of regimen of treatments with total blood volume loss at 48 h postoperative, adjusted for covariates.

Variable	Coeficient	Confidence Interval (95%)	*p*-Value
Group of treatment:			
PBM	(base)		
PBM + Topical	−238.05	(−398.02, −78.08)	**0.004**
PBM + IV	−240.54	(−374.97, −106.11)	**0.000**
PBM + 2IV	−293.15	(−435.00, −151.31)	**0.000**
PBM + Combined	−294.65	(−441.80, −147.50)	**0.000**
Gender:			
Female	(base)		
Male	80.79	(1.39, 160.19)	**0.046**
Age (yo)	3.15	(0.12, 6.18)	**0.042**
Body Mass Index:			
Healthy weight	(base)		
Overweight	121.50	(39.57, 203.43)	**0.004**
Obesity	286.17	(177.63, 394.72)	**0.000**
ASA:			
I	(base)		
II	−162.87	(−313.00, −12.74)	**0.033**
III	−105.77	(−277.51, 65.97)	0.227
Pre−operative Haemoglobin classification:			
<12 mg/dL	(base)		
12–14 g/dL	71.65	(−144.53, 287.83)	0.516
>14 g/dL	208.50	(7.11, 409.90)	**0.042**
Constant	334.88	(60.79, 608.96)	**0.017**

GLM Model: variance function: V(u) = 1 [Gaussian]; link function: g(u) = u [Identity]; Pearson = 8054295.62; (1/df) Pearson = 57530.68; AIC = 13.87; BIC = 8053591. Predicted model: y = 334.88 + β Group of treatment + βAge * (Age) + βBMI + βASA + βHb preOp.

**Table 4 jcm-09-01952-t004:** Comparison of blood loss of published articles with this study data.

Article	Blood Saving Protocol	TXA Scheme	Dose	*n*	TBV Lossat 48 h	SD	Delta
Our study	PBM	none	__	29	738.3	367.4	
Pérez-Jimeno 2018	PBM-like	none	__	129	728 ^a^	252	−10.3 ^ns^
Yi 2016		Placebo	__	50	1221	386	482.7 ***
Wei 2014		Placebo	__	100	1364	279	625.7 ***
Fraval 2017		Placebo	__	51	1394	426	655.7 ***
Barrachina 2016		Placebo	__	37	2215	1136	1476.7 ***
Our study	PBM	2 TXA IV	20 mg/kg	41	487.0	221.4	
Fraval 2017		2 TXA IV	15 mg/kg	50	1084	440	597.0 ***
Barrachina 2016		2 TXA IV	10 mg/kg	36	1308	641	821.0 ***
Our study	PBM	TXA TOP	3 g	18	537.5	330.1	
Pérez-Jimeno 2018	PBM-like	TXA TOP	2 g	125	539 ^a^	243	1.5 ^ns^
Xie 2016		TXA TOP	3 g	70	905	238	367.5 ***
Wei 2014		TXA TOP	3 g	102	963	421	425.5 ***
Gomez 2019		TXA IA	2 g	47	1280.0	352.6	742.5 ***
Our study	PBM	TXA IV	20 mg/kg	42	513.7	265.3	
Gulabi 2019		TXA IV	2 g	26	848.81	224.1	335.1 ***
Xie 2016		TXA IV	1.5 g	70	878	686	364.3 ***
Wei 2014		TXA IV	3 g	101	959	422	445.3 ***
Yi 2016		TXA IV	15 mg/kg	50	1003	367	489.3 ***
Borrachina 2016		TXA IV	15 mg/kg	35	1377	689	863.3 ***
Gomez 2019		TXA IV	15 mg/kg	31	1515.4	499.3	1001.7 ***
Our study	PBM	3 TXA IV + TXA TOP	20 mg/kg	23	434.2	155.2	
Gulabi 2019		TXA IV + TOP	3 g	22	772.22	322.07	338.0 ***
Xie 2016		TXA IV + TOP	3 g	70	777	189	342.8 ***
Yi 2016		TXA IV + TXA IA	3 g	50	836	344	401.8 ***

^a^ Data at 24 h postoperative. *t*-Test: *** < 0.001; ^ns^ = not significant.

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
