# Peer review of "Defining the Most Effective Patient Blood Management Combined with Tranexamic Acid Regime in Primary Uncemented Total Hip Replacement Surgeryâ€"

_jcm, 2020, doi:10.3390/jcm9061952_

Round 1
Reviewer 1 Report
We suggest that the manuscript is rejected in its current form and only be reconsidered if the authors can establish that they can answer the question posed in the title, what is the optimal TXA dosing regimen in total hip replacement surgery.
Major Comments:
- The perceived goal of this manuscript was to fill a perceive gap in knowledge surrounding the comparative efficacy of different TXA administration regimes relative to TBV, Hct, and Hb levels. This retrospective study was conducted at a single hospital that has reported excellent outcome upon introduction of a PBM protocol. Based on the information provided, it is suggested in the introduction that application of the PBM protocol alone reduced the need for transfusion in THR to 0% (line 43). Is this correct? It is difficult to interoperate the conclusion brought by the authors that “doses of 20mg/kg TXA at induction followed by 20mg/kg infusion for duration of the procedure are sufficient to consolidate a 0% transfusion rate (line 295). Would this not be possible with the PBM alone based on earlier statements?
Thus, the clinical conclusion and relative patient outcomes presented do not fully illustrate the recommendation to utilize TXA over PBM alone- though this reviewer would agree TXA administration is helpful! A more clear conclusion relative to estimated TBV loss and patient length of stay would be helpful to assessing the clinical improvement upon TXA usage, rather than transfusion rate (which was already 0%).
- From the outset, it is difficult to decipher the objective of this study. The title, which is awkwardly stated as a question, suggests that the efficacy of different TXA dosing regimens will be investigated. The introduction states that the “primary objective of this study is to compare uncemented THR post-operative blood loss at 48 hours in different PBM and TXA administration groups”. Again, suggesting that different TXA dosing regimens or routes of administration will be compared. Yet the hypothesis states that “the application of TXA combined with PBM for uncemented THR would reduce the TBV compared with PBM alone” with no mention of expected results in terms of significant differences in efficacy between the “TXA administration groups”. An investigation of the efficacy of TXA administration alone, without considering differences between dosing regimens, is not novel and doesn’t advance the field of orthopaedic surgery.
- Currently patients who did not have 48-hour laboratory values were excluded from this study. This may be removing a subset of patients who did very well and were discharged prior to this mark. Therefore, this selection bias may hide an important consideration and that is average patient length of stay. Did patients with different treatment regimes have different LOS? Was the percentage of patients discharged by 48hrs different between groups? This would be clinically very informative and a much stronger clinical output value then transfusion %.
Given that blood loss has been estimated at earlier time points as noted in the manuscript, I would prefer to see values compared at 24 hrs post surgery and include comparisons of LOS between patients (either as raw day values or as the % of patients discharged).
Aligning with this, were any patients readmitted? While not powered to assess complications, please note the percentage of patients that necessitated readmission.
- What was the clinical intuition for selection of TXA regime, beyond topical vs IV? This is not well described in the methods or discussion and may indicate a variance in patient population. Please include analysis of patient comorbidities between groups, such as with a Charlson comorbidity index (CCI). CCI has been shown in other larger studies to be predictive of patient outcomes and hospital readmission following TJAs.
- Please expand on the limitations of estimating the total blood volume loss in the discussion. Additionally, there are many techniques for estimating blood volume loss with TJAs, which is not noted. The formula presented is not referenced and is unclear which method is being used since transfusion is not included. Please review the equation included, provide a validated reference for this measure, and include comments on why for example, the Mercuriali method was not utilized (https://www.ncbi.nlm.nih.gov/pmc/articles/PMC3609988/)
- Please include a power calculation- it is noted that this study was not powered to detect changes in complications, given the relatively few complications patients experienced. Is the study appropriately powered to indicate that there is no statistical difference between the groups- in particular groups 2-4. This is essential to interpreting the data and the conclusions drawn by the author.
Moreover, it is critical that the authors establish their study is powered to detect significant differences in Hb, Hct, or blood loss amongst these groups. The authors have not established that their study is capable of answering the novel question within their cohort.
Please include a P value in table 1 for the proportion of obese patients- this appears to be different amongst groups and potentially impactful to the data analysis-- Given that elevated BMI is significantly associated with an increase estimated blood loss in total joint replacement, obese patients have been found to actually have lower rates of blood transfusion and overall significantly smaller percentages of estimated blood volume loss. This needs to be discussed and the authors should include a line in Table 2 indicating the % total blood loss per patient, rather then the raw values of estimated total blood loss. Furthermore, the comment in the abstract about BMI driving TBV is not novel and potentially misleading as a raw value..
- At no point in this study is there a functional measure of fibrinolysis. While noted in the discussion that patients with TJA can experience a hyperfibrinolytic state (more references are needed), this study does not consider measures of such a state (ie d-dime) to evaluate if the TXA administered is truly efficacious. This is a massive limitation to retrospective studies attempting to validate TXA administration strategies and must be commented on in the discussion. Furthermore, at no point is the dosage of TXA (20mg/kg) varied in this study- which is understandable given the retrospective nature of the study. BUT this does mean the title is misleading and needs to be changed- because the dosage is not altered, the administration method is.
Furthermore, the title should not include the word optimal- since there is no molecular comparison to illustrate that one option is better than another and no variance in dosing. Rather this study suggests that they are all the same which the title should reflect. Further, the fact that dosage is not considered in this manuscript needs to be addressed as a limitation.
- Several studies were even cited in the introduction that have already investigated this question and many more exist in the literature. Nevertheless, the authors proceed to demonstrate that reductions in Hb and Hct “did not show differences among the groups with TXA” and that “a similar volume of blood loss was found among the groups that received the PBM program and at least one dose of ATX”. These statements of differences and similar volumes should be qualified as significant or non-significant.
- Figure 1: please add statistical significance to the reported values. This reviewer finds the stacked columns difficult to interpolate and would consider side-by-side- columns. Given the robustness of the data at 24 hrs, this reviewer continues to recommend removing the exclusion criteria of data at 48 hrs.
Figure 2: please correct the Y axis in figure 2 to read Total estimated blood volume loss at 48 hours. Please change this to be 24 hour data and 48 hour data, or just 24 hrs. Also consider reporting a % of blood loss since raw values may mask the true patient impact.
Figure 3: Please provide a y axis label. Please provide statistical comparison between PBM and the other groups, as well as between the TXA administration groups. If the authors want to conclude that there is no difference, they must show that the study is appropriately powered, and must include statistic comparisons across these groups.
Table 3: requires significantly more discussion in the result to interoperate the data. Based on the clinical audience the utility of such a table is lacking without proper discussion and interpretation. I would consider moving this table to supplemental or expanding on its interpretation greatly.
- There are many misspellings of TXA throughout manuscript. Please review the entire manuscript carefully for such errors.
- Much of the discussion comments regarding patient care (line 206-212, 224-229,) can be simplified and added to the methods section to provide more streamlined discussion. Furthermore, many of the discussion statements require references (lines 229-234). Finally, the comment on easy of protocol implementation- (line 263), specifically which protocol is being referred to given that 5 cohorts are examined. How is the author quantifying this statement or is this an observation without data to support it?
- Table 4: Definitions need in figure legend along with notes on statistical test used.
Reviewer 2 Report
Thank you for giving me the opportunity to review this manuscript on patient blood management and tranexamic acid administration in THR patients. I enjoyed reading this manuscript. The authors have made a great effort to collect and analyze this quite interesting data. In my opinion, the subject matter of this research is quite important, gaining more and more popularity nowadays in adult reconstruction surgery. The methodology of the study is pertinent and the provided results consist an important input to the related literature. The text is well written. Tables, charts and the references used are appropriate.
I have only some minor comments:
-The abstract of this study is difficult to read. I understand that the authors need to provide enough information in limited sentences, however I feel that the results and the main message are difficult to be identified by the general reader. I would suggest the authors to modify it accordingly, as a clear abstract attracts the attention to keep reading the main manuscript.
-Similarly, the introduction section is very exhaustive. I would recommend the authors to provide a more succinct introduction section, and be descriptive to the discussion section.
-In the first two paragraphs of the results section demographics are described. In my opinion this should be moved to the materials and methods section, as it represents the materials of the study and not the results. Eventually this info should be incorporated with the text between lines 100-111.
-The conclusion section should be limited in one or two final statements. Further explanation of the results is not needed at this section, as everything is already analyzed in the text before.
-Informed consent description and IRB approval should be moved to the materials and methods section.
Otherwise, I felicitate the authors for their nice work and I recommend publication of their manuscript after these minor changes.
Reviewer 3 Report
Dear Authors,
perioperative blood loss is a significant concern for patients undergoing total joint arthroplasty. A growing body of evidence has shown tranexamic acid (TXA) to be effective in decreasing perioperative blood loss and transfusion requirements in both primary and revision hip and knee arthroplasty. Both topical and intravenous administration of TXA, in a variety of dosing regimens, has proven effective.
Within the last years, numerous studies have shown TXA to be beneficial, but extensive variability exists with regard to dosage, timing of TXA administration, and the number of doses required. Additional heterogeneity exists among study designs, and small patient numbers further limit many studies. As a result, drawing clear conclusions regarding appropriate dosage and timing of administration are difficult. These studies conclude, that further investigation is required to determine the optimal dose and dosing regimens.
You performed a retrospectively analysed cohort study of prospectively collected data from uncemented THR patients operated at the Hospital Universitario La Paz-Cantoblanco from January of 2007 through December of 2015. The primary objective of this study was to compare the uncemented THR post-operative blood loss at 48 hours in the different PBM and TXA administration groups. The multivariate regression model confirmed a significant decrease of TBV in all groups with TXA compared with PBM only group. The ideal regime to achieve our transfusion rate of 0% and the least blood loss required PBM and one dose IV TXA in uncemented THR. Additional doses of TXA did not add benefit.
Although numerous studies have already shown TXA to be beneficial in reducing the rate of blood loss and the transfusion requirements associated with primary THA, the dosage and administration are still under debate.
Therefore your study is offering some new information on that topic to achieve an optimal Patient Blood Management (PBM)using tranexamic acid in THA with a considerable interest to the readers.
This paper is offering a good quality of presentation, English language and style are fine and only minor spell check is required!
Best Regards
Round 2
Reviewer 1 Report
Overall, the authors did a good job refining the scope and focus of this manuscript.
Clarification that this study was focused on examining the route and timing of administration is essential. Given that dose of TXA was not examined in this manuscript, we would advise adding a line to the abstract indicating that “this study aimed to examine the effectiveness of timing and route of administration of TXA, in combination with PBM, at reducing TBV lost following total hip replacement surgery.” By adding a clearer directive, this will aid future readers to the important points of the study—much like the last paragraph in the introduction; this was a very clear statement of the goals of this work.
Aligning with the goal of the study, please remove the term "optimal" from the title. This is overstepping the study design since dose was not a variable in this study. Therefore, it is not possible to state that this protocol is “optimal” because variables are yet to be examined.
Thank you for clarifying your patient population and extending your data analysis to include power calculations- this greatly improves the interpretation of the data. The addition of LOS is very helpful for interpreting the larger value of this protocol – in addition to improving TBV loss in patients, by having the TXA as multiple times of administration, you are also reducing the LOS and therefore hospital cost. In the US health care system this is a very important point and will be of interest to readers.
For the figures, the data is much improved. However, it would be great to have Y axis labeled appropriately on the graph rather than as a header as currently presented. When prepared for publication, these should be moved/added.
Line 281, when discussing obesity- please also add that an alternative DOSE of TXA could be utilized, in addition to more frequent dosing. This is a nice opportunity to add in the point that future studies considering variation in TXA dose, tailored to the patient, may be necessary in addition to improving the administration timing and route.
Author Response
Overall, the authors did a good job refining the scope and focus of this manuscript.
Clarification that this study was focused on examining the route and timing of administration is essential. Given that dose of TXA was not examined in this manuscript, we would advise adding a line to the abstract indicating that “this study aimed to examine the effectiveness of timing and route of administration of TXA, in combination with PBM, at reducing TBV lost following total hip replacement surgery.” By adding a clearer directive, this will aid future readers to the important points of the study—much like the last paragraph in the introduction; this was a very clear statement of the goals of this work.
Aligning with the goal of the study, please remove the term "optimal" from the title. This is overstepping the study design since dose was not a variable in this study. Therefore, it is not possible to state that this protocol is “optimal” because variables are yet to be examined.
Thank you for clarifying your patient population and extending your data analysis to include power calculations- this greatly improves the interpretation of the data. The addition of LOS is very helpful for interpreting the larger value of this protocol – in addition to improving TBV loss in patients, by having the TXA as multiple times of administration, you are also reducing the LOS and therefore hospital cost. In the US health care system this is a very important point and will be of interest to readers.
For the figures, the data is much improved. However, it would be great to have Y axis labeled appropriately on the graph rather than as a header as currently presented. When prepared for publication, these should be moved/added.
Line 281, when discussing obesity- please also add that an alternative DOSE of TXA could be utilized, in addition to more frequent dosing. This is a nice opportunity to add in the point that future studies considering variation in TXA dose, tailored to the patient, may be necessary in addition to improving the administration timing and route.
PLEASE SEE ATTACHED DOCUMENT
